# Neural Plasticity Changes Induced by Motor Robotic Rehabilitation in Stroke Patients: The Contribution of Functional Neuroimaging

**DOI:** 10.3390/bioengineering10080990

**Published:** 2023-08-21

**Authors:** Lilla Bonanno, Antonio Cannuli, Loris Pignolo, Silvia Marino, Angelo Quartarone, Rocco Salvatore Calabrò, Antonio Cerasa

**Affiliations:** 1IRCCS Centro Neurolesi Bonino Pulejo, 98123 Messina, Italy; lilla.bonanno@irccsme.it (L.B.); antonio.cannuli@irccsme.it (A.C.); silvia.marino@irccsme.it (S.M.); aquartar65@gmail.com (A.Q.); 2S’Anna Institute, 88900 Crotone, Italy; l.pignolo@isakr.it; 3Institute for Biomedical Research and Innovation (IRIB), National Research Council of Italy (CNR), 98164 Messina, Italy; 4Translational Pharmacology, Department of Pharmacy, Health and Nutritional Sciences, University of Calabria, 87036 Rende, Italy

**Keywords:** robotic neurorehabilitation, motor recovery, fMRI, fNIRS, stroke

## Abstract

Robotic rehabilitation is one of the most advanced treatments helping people with stroke to faster recovery from motor deficits. The clinical impact of this type of treatment has been widely defined and established using clinical scales. The neurofunctional indicators of motor recovery following conventional rehabilitation treatments have already been identified by previous meta-analytic investigations. However, a clear definition of the neural correlates associated with robotic neurorehabilitation treatment has never been performed. This systematic review assesses the neurofunctional correlates (fMRI, fNIRS) of cutting-edge robotic therapies in enhancing motor recovery of stroke populations in accordance with PRISMA standards. A total of 7, of the initial yield of 150 articles, have been included in this review. Lessons from these studies suggest that neural plasticity within the ipsilateral primary motor cortex, the contralateral sensorimotor cortex, and the premotor cortices are more sensitive to compensation strategies reflecting upper and lower limbs’ motor recovery despite the high heterogeneity in robotic devices, clinical status, and neuroimaging procedures. Unfortunately, the paucity of RCT studies prevents us from understanding the neurobiological differences induced by robotic devices with respect to traditional rehabilitation approaches. Despite this technology dating to the early 1990s, there is a need to translate more functional neuroimaging markers in clinical settings since they provide a unique opportunity to examine, in-depth, the brain plasticity changes induced by robotic rehabilitation.

## 1. Robotic Neurorehabilitation in Stroke Patients

One of the main global causes of mortality and long-term impairment is stroke. It occurs when blood flow to the brain is disrupted, causing brain damage involving motor, sensory, and cognitive functions negatively affecting quality of life. Rehabilitation is a critical component of stroke management and can help patients recover lost abilities and improve their quality of life [1]. In recent years, robotic neurorehabilitation has been demonstrated to be effective in the recovery of motor functions in stroke patients [2].

Robotic neurorehabilitation involves the use of robotic devices to provide repetitive, high-intensity and task-oriented training to stroke patients. These devices are designed to assist movements, depending on the patient’s needs, providing real-time feedback on performance. Promoting neuroplasticity, the brain’s capacity to restructure and generate new neural connections in response to injury by providing targeted and intensive training, is the aim of robotic neurorehabilitation. Although several robotic devices have been designed and commercialized for patients with stroke, they can easily be classified into two main categories: exoskeletons and end-effectors. Exoskeletons are commonly used in patients with more severe deficits, including those with complete hemiplegia, whereas individuals with mild to moderate deficits may better benefit from the functional challenges offered by end-effectors [3].

Lokomat and Armeo-Power are examples of lower and upper limbs, respectively, whilst the Amadeo and Geo-system are examples of hand and gait end-effectors (Figure 1).

Moreover, besides these fixed devices, patients with stroke may be trained using overground exoskeletons, including the Ekso-GT, and other wearables supporting different parts of the body.

Current literature suggests that robotic neurorehabilitation can be effective only when these advanced tools are added to standard rehabilitation for stroke patients and in most severely affected patients, as well as within the first 3 months following stroke. In a published systematic review [4], the authors analyzed 17 randomized controlled trials that investigated the effectiveness of robotic-assisted upper limb training for stroke patients. They found that robotic therapy improved upper limb function compared with conventional therapy alone, and that the benefits persisted for up to 6 months after treatment. Moreover, it seems that chronic patients benefit the most from advanced training. The data were then confirmed by a recent systematic review [5] and meta-analysis (2021) analyzing 41 randomized controlled trials that investigated the effectiveness of robotic-assisted upper limb training in mild to moderate arm impairment. The authors found that robotic therapy was more effective than conventional therapy in improving upper limb function and that the benefits persisted for up to 12 months after treatment. They also found that higher-intensity robotic therapy led to greater improvements in upper limb function than lower-intensity therapy. However, more high-quality research is needed to determine the optimal type and duration of robotic therapy.

Concerning lower limbs, a Cochrane review published in 2017 [6] analyzed 36 randomized controlled trials that investigated the effectiveness of robotic-assisted gait training for stroke patients. The authors found that electromedical devices improved walking speed and distance compared with conventional therapy, but the effect sizes were small. These findings were confirmed by the same group in 2020 [7], analyzing 62 randomized controlled trials involving 2440 participants. It was indeed shown that robotic therapy improved walking speed, endurance, and balance, compared with conventional therapy, and that the benefits persisted for up to 6 months after treatment. However, the authors noted that more high-quality research is needed to determine the optimal type and duration of robotic therapy as well as the specific device to use. To this aim, an interesting work by Morone et al. [8] suggested that the different robotic devices may be properly used according to the functional ambulation scale scores of the stroke patients. In particular, patients more severely affected may benefit from stationary exoskeletons (including the Lokomat), whereas those who can walk with minor aid or autonomously should receive only conventional gait training.

Therefore, from the current literature, it seems that the combination of robotic therapy and standard rehabilitation is more effective than standard rehabilitation alone in functional recovery of stroke patients. However, more high-quality research is needed to determine the optimal type, duration, and intensity of robotic therapy, as well as its long-term benefits on functional outcomes.

## 2. Determining the Functional Impact of Robotic Neurorehabilitation

The efficacy of robotic neurorehabilitation is generally measured, as every kind of treatment, with well-known and standardized clinical scales, such as the Fugl-Meyer Assessment (FMA), the Functional Ambulation Scale (FAC), the 10 m and 6 min walking tests, etc., FMA is the most widely used assessment tool for evaluating motor impairment and recovery in stroke patients. It measures impairment in the upper and lower limbs, as well as the trunk, and assesses movement quality, coordination, and reflex activity. The FMA is often used to evaluate the impact of robotic-assisted therapy on motor function in stroke patients for research purposes. However, its validity and use in clinical practice to investigate robotics after-effects is conceivable.

Clinical scales, on the other hand, have a low sensitivity to assess the true neurobiological effects of an advanced neuro treatment. The most efficient technique to assess the impact and effect of robotic neurorehabilitation thoroughly and more objectively in stroke patients is to use neuroimaging tools (such as fMRI and fNIRS) [9].

For the past 30 years, movement disorders [10] as well as the mechanisms underpinning movement regulation [11] have been studied using functional magnetic resonance imaging (fMRI). A very high spatial resolution indirect measure of the functional activity of the human brain is provided by the blood oxygenation level-dependent (BOLD) signal, which depends on changes in deoxy-hemoglobin (deoxyHb) concentrations [12]. Recently, the focus has switched to investigating the characteristics of distributed networks [13]. This shift in viewpoint is largely attributable to the development of the network science, which gave rise to a framework for mathematically expressing and analyzing high dimensional datasets [14]. A “connectomics revolution” [15] was sparked by these two things, as well as data-sharing programs and access to powerful computers, and led to the explicit study of the structure and function of the brain from a network perspective in neuroscience.

However, this method is severely constrained in terms of motion artifacts, and only very small motions are permitted inside the scanner. To work around these restrictions, several ways have been tried, mainly by researchers interested in evaluating the neurofunctional correlations of lower limb functions. For instance, numerous neuroimaging studies on motor imagery have been carried out [16], although imaging varies from subject to subject, and real walking engages different brain networks than imagined walking [17,18]. There have been several attempts to enable a nearly actual walking sequence while using fMRI [19], while other writers have recommended using virtual reality [20]. Use of surrogate activities in the scanner as a stand-in for walking tasks provides additional chances to study the mechanisms sustaining movement control [21]. Consequently, there has not been an ecological technique for noninvasively assessing the neurophysiological components of walking processes in motor disorders.

In order to overcome these particular limitations, functional near-infrared spectroscopy (fNIRS) has increasingly been proposed as an alternative crucial research technique [22,23,24,25,26]. A noninvasive optical imaging method called fNIRS measures the hemodynamic response to infer the underlying brain activity, akin to fMRI. Oxyhemoglobin (oxyHb) and deoxyHb, which have various absorption spectra depending on the photon’s wavelength, and transmit near-infrared light (650–1000 nm) into scattering tissues, are the two main chromophores in the brain [27]. Typically, a fNIRS device includes a light source attached to the participant’s head using fiber-optic bundles or light-emitting diodes, as well as a detector that gathers the light after it has been reflected off the tissue. The optical density of the photons that are recorded by detectors varies depending on how biological tissues absorb light. By combining many wavelengths with the modified Beer–Lambert rule, it is feasible to determine how the concentrations of oxyHb and deoxyHb have changed [28]. fNIRS has several undeniable advantages over fMRI, its major competitor. The fNIRS device offers the following benefits because it measures changes in both oxyHb and deoxyHb concentration with high temporal resolution (up to milliseconds), namely, it: (a) does not impose immobility restrictions [29], (b) is portable [30], (c) allows recording while actually walking [31], (d) allows long-lasting recordings, (e) allows the study of brain activity during sleep [32], and (f) enables the development of a more thorough understanding of the neurovascular coupling. The analysis of a hemodynamic response, whose dynamics take at least 3 to 5 s, typically does not require high temporal resolution, but it can be useful for the study of transient hemodynamic activity [33]. Comparing fNIRS to fMRI, its primary disadvantages are its inferior spatial resolution (a few millimeters beneath the head) and insensitivity to subcortical areas [34,35]. This might be viewed as a minor drawback, though, as a substantial body of research suggests that walking involves cortical mechanisms [36], the motor system is organized across broad brain regions [37], and the cerebral cortex mirrors the function of subcortical structures [38]. Overall, fNIRS has been demonstrated to be a valid tool for assessing in vivo the cortical brain activity linked to the activity of the upper and lower limbs.

For this reason, in order to determine whether there is sufficient evidence to define distinct brain functional patterns connected to motor recovery driven by neurorobotic rehabilitation, we set out to provide a review of recent works. This review discusses the current methodology and technology used for robotic neurorehabilitation. Despite the numerous studies showing that robotic neurorehabilitation is beneficial for recovering motor function, few studies have demonstrated the presence of long-term and persistent neural plasticity rearrangements.

Basically, it has been shown that standard rehabilitation protocol induced adaptive neural changes in the medial premotor and primary motor cortex. Favre et al. [39] carried out an activation likelihood estimate meta-analysis of fMRI studies looking at upper limb movement-related brain activity after stroke to carefully assess the neural plasticity changes associated with good or poor outcome. Patients displayed elevated activation likelihood estimation values in the controlesional primary motor cortex, although these values gradually decreased over time, and were unrelated to motor outcome. With the restoration of the usual interhemispheric balance, the observed activity variations were consistent. In contrast, positive outcomes were associated with activation probability estimation values in the primary motor cortex and medial premotor, a rearrangement that might be caused by vicarious processes linked to ventral activity shifts from BA4a to 4p.

The goal of the current systematic review is to pursue this important area of research by determining whether there is sufficient evidence to support the existence of neurofunctional biomarkers that may be used to evaluate the effectiveness of robotic therapy in stroke patients.

## 3. Methods

This systematic review was conducted on papers published on the use of functional neuroimaging tools (fMRI, fNRIS) to determine the presence of neural plasticity changes in stroke patients after intense robotic neurorehabilitation of motor disorders. The PICOS approach was utilized to identify the studies to be included in the review, which was structured and registered in accordance with the PRISMA statement [40]. Criteria for including or excluding papers were determined a priori. Using electronic bibliographic databases, including PubMed, PsycINFO, PEDro, Scopus, and Cochrane Library, articles published between January 2007 and January 2022 were evaluated (Figure 2). MeSH and “text words” were employed as keywords to enhance the search technique. The search terms were concatenated in an advanced query using Boolean operators as follows: “Task fMRI” OR “NIRS” AND “Stroke” AND (“rehabilitation” OR “robotic” OR “robotic rehabilitation”). The search method was tested and used in PsycINFO, and then it was modified for usage in every other electronic database. Duplicate studies were removed when the results were downloaded into the bibliographic management application EndNote X7. Due to the large number of studies found, only one author (L.B.) evaluated each study’s title and abstract to determine whether it met the requirements for inclusion in the review. A second author (A.C.) verified each study’s eligibility after the final pool of included research was determined. The full-text papers under examination had their reference lists reviewed for any further pertinent works.

Papers were included or excluded based on predetermined criteria. Only papers that met the following criteria were given consideration for inclusion: (a) full-text English publication in a peer-reviewed journal, and (b) post-rehabilitation use of fMRI/fNRIS. Articles were excluded if they: (a) lacked English language content, and (b) were dissertations, book chapters, or conference papers that had not yet been published. As we were interested in characterizing the long-term brain plasticity changes associated with intensive rehabilitation sessions, we did not include articles addressing the neural activity during fMRI tasks executed with robotic devices.

The information gathered from each article was divided into categories such as first author, year of publication, neuroimaging modality, cohort size, intervention modalities, experimental methods, outcomes (physical and psychological), and main results.

The methodological quality and bias risk of each study were evaluated using the Quality Assessment of Diagnostic Accuracy Studies (QUADAS) methodology. Through this quality evaluation, studies could be categorized as having a low, high, or unknown risk of bias.

## 4. Results

Identification, screening, eligibility, and inclusion were the four stages of the method used to choose the studies for this qualitative review, as shown in Figure 1. After the initial review, 55 were still present. Forty-one were not included in the second phase because they did not meet the requirements. Finally, seven articles were included in this review, describing experiments where stroke patients and healthy controls underwent fMRI/fNIRS pre and post robotic rehabilitation. The trial size ranged from 8 to 60 participants. The mean age of patients recruited ranged between 49 and 57 years.

In this section, the seven studies that met the inclusion/exclusion criteria are summarized. The pertinent findings are summarized in Table 1 to allow easier comparison.

### 4.1. Robotic Devices

The main robotic devices employed in these studies were:(1)BCI-guided robot-assisted (exoskeleton) for upper-limb training [40]: This is a rehabilitation system for practicing hand movements. It includes a number of finger assemblies that are operationally connected to a platform: each finger assembly has a motor for a metacarpophalangeal joint having a proximal rail guide operationally connected, and an intermediate assembly for a proximal interphalangeal joint having an intermediate rail. The alignment of the knuckle joint indicators allows motion of the finger to be controlled and maintains rotational axes of the finger about each virtual center when the proximal and intermediate follower assemblies are actuated by the motor. A knuckle joint indicator of the proximal rail guide corresponds to a first virtual center, and a knuckle joint indicator of the intermediate rail guide corresponds to a second virtual center.(2)Hand-induced robotic device (MR_CHIROD) [41]. This is a redesigned robotic hand device that allows participant grip and release of a handle in response to a variable resistance force while watching an oscillating visual stimulus. The MR_CHIROD v3 (MR-compatible hand-induced robotic device) is a device that displays customizable forces for grasping and releasing motions while simultaneously measuring and recording applied force, grip displacement, and timestamps for each data point [42].(3)Technology-robot-assisted virtual rehabilitation adaptive training system for upper limbs (NJIT-RAVR) [43,44]. This consists of the Haptic Master, six degrees of freedom, admittance-driven robot, and a collection of rehabilitative simulations that supply the Haptic Master with adaptive algorithms, enabling interaction with detailed virtual worlds. The movement arm can serve as an interface between the participants and the virtual worlds, by measuring the external force applied by the user to the robot, together with end-point location and velocity, in 3D in real time at a rate of up to 1000 Hz. When used as the end effector, the ring gimbal adds the ability to rotate the forearm and counts three extra degrees of freedom. The robot actively generates and records the force that aids or opposes forearm rotation (i.e., roll). Pitch and yaw angles, on the other hand, are passively recorded. The robot can be programmed to provide haptic effects such as springs, dampers, and constant global forces, due to the haptic master application programming interface.(4)Hand–wrist assistive rehabilitation device (HWARD) [45]. This pneumatically operated, three-degrees-of-freedom tool supports the hand’s grab and release motions. The three angles are the wrist’s flexion and extension, the thumb’s flexion and extension at the MCP joint, and the four fingers together around the MCP joint. The individual is sitting and looking at a computer screen. Three gentle straps hold the hand to the robot’s mechanism, while a padded splint mounted to the platform’s surface holds the forearm in place. The palmar hand is not constrained, allowing genuine things to be placed into a gripping hand. The movement of the robot’s joints and, consequently, the movement of the subject’s limbs when coupled to the robot are measured by joint angle sensors in the robot. This feature allows the operation of a virtual hand on a computer screen, by using the subject’s hand in real-time virtual reality. When the robot is not actively assisting individuals, it can be back-driven, allowing subjects to move freely.(5)BCI-robot training system of upper limbs (RHB-III) [46]. This consists of a robotic exoskeleton used to help the paretic hand perform the real movement of grasping/opening tasks. Participants were told to view the motions in the video and follow instructions to visualize doing the identical motion with their paretic hand. The calculation of the *mu* event-related desynchronization score was performed based on the real-time EEG readings. When the score exceeded 60, the robot was activated and helped the paretic hand complete the grasp/open job for the following three seconds. However, if the mu event-related desynchronization score was less than 60, the trial was deemed unsuccessful and the robot was not triggered to move. Motor imagery instruction was promoted until the video screen showed successful or unsuccessful detection. If motor imagery was correctly identified, the robot gave visual and movement feedback by actually moving the paretic hand.(6)MorningWalk^®^ system for gait recovery [47]. This was created in 2014 by CUREXO Inc. in Korea for the rehabilitation of individuals with gait disorders. It has a saddle that can bear the weight of the sufferer. Virtual reality augmentation software enables interactive training. The gadget allows for complicated treatment protocol designs by allowing users to apply free-walking programs with several parameters for walking speed, stride length, and different walking motions, such as flat ground and stair climbing.

### 4.2. Risk of Bias within Studies

Following QUADAS criteria, the vast majority of studies achieved a low risk of bias, and low concerns regarding applicability [42,43,45,46]. The other papers showed a high risk of bias regarding the appropriateness of reference standard and index test [41,44,47] (Figure 3).

### 4.3. fMRI Studies

#### 4.3.1. Work by Yuan et al., 2020 [41]

In this study, Yuan and colleagues were interested in evaluating the functional reorganization and long-term structural changes induced by BCI-guided robot-assisted training. Neuroimaging and clinical assessment were performed before, immediately after, and six months after a 20-session BCI-assisted training protocol.

While functional connectivity (FC) analysis was carried out using a seed-based technique with four seeds placed at various motor-related areas in the ipsilesional hemisphere, structural integrity was assessed using diffusion tensor imaging (DTI). The lateralization index changes of activity in the contralesional motor cortex against the ipsilesional motor cortex were also assessed when a motor imaging task was performed with the paretic hand. Additionally, the association between motor function alterations and FC, as well as the integrity of the ipsilesional corticospinal tract (CST), were also evaluated.

Fourteen chronic stroke subjects (mean age 54 ± 8 years), with left (*n* = 5) or right (*n* = 9) hemisphere impairment, were enrolled considering first-ever stroke. The motor function of the paretic upper limbs was assessed using the Fugl-Meyer Assessment (FMA) before, immediately after, and six months after the intervention, respectively. The BCI-guided training task consisted of imagining holding or releasing a cup in accordance with the instructions shown on the monitor. An exoskeleton robot hand [48] was utilized during the task to help the paretic hand with grasping and opening. Each subject’s EEG data were simultaneously acquired using portable signal acquisition equipment with 16 electrodes covering the motor-related areas. Before and after the neurorehabilitation program, MRI scans were obtained.

FC analysis revealed significant interhemispheric functional reorganizations. Increased functional connectivity between the ipsilateral primary motor cortex and the contralateral lateral premotor cortex was detected immediately after neurorehabilitation, whereas increased connectivity with the SMA emerged after 6 months. When the seed was positioned into the SMA, a significantly increased FC was detected in the superior parietal cortex either after rehabilitation or at 6-month follow-up revaluation. All overall patterns of FC changes, as well as the structural integrity of the CST, correlated with motor improvements as measured by the FM test. Finally, after training, the ipsilesional hemisphere was comparatively more active, according to a task-based LI analysis.

In conclusion, this study showed that training with a robot supported by a BCI system resulted in long-term functional and structural remodeling.

#### 4.3.2. Astrakas et al., 2021 [42]

In this neuroimaging study, the authors investigated 8 stroke patients (mean age 49.9 ± 12.7 years), compared with demographically-matched 13 healthy controls. Neurorehabilitation consisted of robot-assisted rehabilitation therapy. Over the course of ten weeks, patients were trained 45 min per day, three days per week, at home using the third-generation Magnetic Resonance Compatible Hand-Induced Robotic Device (“MR_CHIROD”) and an interactive video game. Their motor function was evaluated prior to training (baseline), in roughly monthly intervals throughout training to track development, and one month after training was finished (follow-up) to gauge perseverance through time.

Five scan sessions were performed: one at baseline, three throughout rehabilitation, and one at the one-month follow-up. The HCs were subjected to a solitary scanning session.

fMRI motor tasks consisted of a simple grip task performed by patients using their paretic hands. The MR_CHIROD device was used for the fMRI motor task, and was attached to the scanner table adjacent to the subject so that the subject’s right-hand palm and fingers could easily manipulate the robotic device’s handle. The MR_CHIROD was compressed and released during the action phase in time with a metronome’s visual signal. The MR_CHIROD device was used to measure each subject’s maximum grip strength prior to scanning by gradually increasing the grasp-resisting force until a complete grasp closure was impossible. Six separate motor sessions were produced by performing the paradigm with both hands while applying resistive forces of 60%, 40%, and 20% of the subject’s maximum grip strength.

A comparison of the MNI coordinates of the peak activation locations within the sensorimotor areas between HCs and stroke patients revealed an average 5.3 mm anterior shift of ipsilesional peak activation for the paretic right hand. The coordinates of the sensorimotor peak activation for the unaffected left hand did not change across groups.

Due to the small sample size, the main limitation of this study was that the authors did not provide a statistical comparison between patients and controls, but they only reported within-group analyses.

#### 4.3.3. Saleh et al., 2017 [43]

In this pilot study, the neural pattern reorganization associated with intensive robot-assisted virtual reality (RAVR) therapy with respect to repetitive task practice (RTP) was compared. Based on measurements of functional and effective connectivity before and after the two different treatments, the fMRI analysis included the magnitude, extent, and relateralization of brain activations. Brain activity and clinical outcome variables were also connected in order to define the significance of brain neuronal remodeling following training.

The individuals were divided into 10 chronic stroke patients (mean age 59.6 ± 10.6 years; two female) who took part in eight 3 h sessions of RAVR therapy, and 9 stroke patients (57 ± 12.8 years, three female) who took part in eight sessions of matching RTP therapy. In the first group, the training involved reaching for and interacting with stationary and moving virtual objects with robotic assistance adapted to the individual needs of each subject. However, the RTP group was sequentially engaged throughout a 2-week training session that included reach-and-grasp exercises and functional activities. The paretic hand was used for the fMRI task, which involved whole-hand finger flexion. Participants were asked to flex their fingers in a virtual reality simulation to meet a set of two visual targets that were tilted at 40 and 80% of their maximum range of motion. Subjects received real-time visual feedback of their movement through data glove input from VR hand models projected on the screen.

Functional neuroimaging metrics (including laterality index and effective connectivity) were evaluated into specific regions of interests, including: contralesional motor cortex, ipsilesional primary somatosensory cortex, ipsilesional ventral premotor area, and ipsilesional supplementary motor area. fMRI data were collected during paretic hand movement before and after training. To ascertain how neurophysiological changes are connected to motor improvement, the Jebsen Taylor Hand Function Test (JTHFT) was employed to evaluate the relationship between changes in fMRI data and functional improvement.

Considering the magnitude and extension of BOLD activity, as well as the functional connectivity after treatments, no significant differences were found when the authors directly compared the two groups, whereas a difference in lateralization index between groups was observed after training. The shift toward greater ipsilesional hemisphere dominance was more pronounced in the RAVR group. The aforementioned change [49] is in line with what is typically seen when the interhemispheric balance is seen over recovery. In particular, compensatory mechanisms that enable the non-lesioned hemisphere to compensate are assumed to be the origin of the early post-stroke change in dominance toward the contralesional hemisphere. Finally, considering the relationship between clinical improvement and functional changes, they discovered a strong association between the bilateral primary sensory regions, the ventral premotor area, and the iMC in the RAVR group but not in the RTP group.

According to the patterns of brain reorganization, the authors concluded that clinical improvement in the RAVR group may have been more associated with the restoration of activity in ipsilesional sensorimotor networks, whereas improvement in the RTP group may have been driven by an adaptive compensatory process in the contralesional hemisphere.

#### 4.3.4. Saleh et al., 2012 [44]

In this pilot study, after two weeks of robot-assisted virtual reality therapy for the paretic upper limb, alterations in brain connection patterns in two right-handed stroke patients were evaluated. Prior to and following intensive training of the affected upper extremity using the New Jersey Institute of Technology robot-assisted virtual rehabilitation (NJIT-RAVR) [50] adaptive training system during visually-guided hand movement, functional MRI data (resting state and task-related fMRI) were collected. An extensive upper extremity training program, known as the NJIT-RAVR process, was used to retrain the subject’s hand and arm coordination as well as reach, grasp, and finger individuation. The patient played video games in a robot-assisted virtual reality environment. This training was conducted three hours per day, four days per week, for two consecutive weeks.

The paretic hand’s movement kinematics during fMRI were compared before and after therapy, and experimental sessions within each scanning day, including detailed comparison of finger movement duration, angular excursion, and angular velocity. Before and after the training session, motor functions, Wolf Motor Function Test (WMFT) and Jebsen Test of Hand Function (JTHF) were evaluated, and it was demonstrated that after 2 weeks of NJIT-RAVR therapy, these functions improved in both subjects, in parallel to the changes in brain activation.

The resting state (rs) and task-related functional connectivity (FC) maps showed a different pattern of reorganization with the ipsilateral primary motor cortex (M1). Specifically, both the rs-FC and the task-related FC revealed a decreased activity between the contralesional sensorimotor primary motor cortex and the M1 for the first patient, whereas an increase in connectivity between bilateral sensorimotor/premotor areas and M1 was detected in the second patient.

In conclusion, given the well-known limitations of this neuroimaging technique, the employment of fMRI on a single subject prevents us from discussing these results as a clinical tool.

#### 4.3.5. Takahashi et al., 2008 [45]

The study intended to develop and assess the clinical effects of a robotic therapy targeting the distal arm to improve motor function in stroke patients. This therapy was thought-based on several principles of motor learning, such as intense, active repetitive movement, sensorimotor integration, and high attentional valence and complexity of the experience incorporating a virtual reality interface and using real objects in a natural context to enhance the motor performance of individuals with hemiparesis and to maximize attention to the task. fMRI brain mapping before and after robotic therapy was performed, evaluating several task parameters. The idea was that while a movement that was not performed would not show an increase in representation area in the stroke-affected primary sensorimotor brain over time, it would in the training of a stroke patient’s distal upper extremity.

Thirteen chronic stroke patients (seven female) ranging in age from 37 to 86 years old (mean disease duration 3 months) were enrolled. Assessment of hand motor function and fMRI evaluation were performed on patients before therapy. Next, participants received treatment for 15 consecutive weeks, followed by a third round of evaluations midway through the course of treatment. Finally, fMRI evaluation was repeated along with a new clinical assessment. One month following the treatment’s finish, a fifth and final evaluation was conducted.

The robotic therapy used the “Hand Wrist Assistive Rehabilitation Device” (HWARD) [51]. This pneumatically operated, three-degrees-of-freedom tool supports the hand’s grab and release motions. The four fingers moving in unison around the metacarpophalangeal joint, the thumb moving in unison with the metacarpophalangeal joint, and the wrist moving in unison with the metacarpophalangeal joint are three degrees. The subject is positioned inside a padded splint that is mounted to the surface of a platform, facing a computer monitor, and their hand fastened to the robot mechanism via three soft straps. Real objects can be put into a grabbing hand using the palmar hand. When a subject’s limbs are coupled to HWARD, joint angle sensors in the apparatus are used to measure the movement of the robot’s joints, or of the subject’s limbs. This feature allows the possibility of operating a virtual hand on a computer screen using the subject’s hand in real-time virtual reality.

Robot-based therapy yielded significant motor recovery as measured by Action Research Arm Test (ARAT) and FM scores. When the grab task from robotic treatment was performed during an fMRI, it revealed increased sensorimotor brain activation during the course of the therapy, in contrast to supination/pronation, which was not a practiced task. As there was no corresponding change in task-related EMG, it was likely that the altered grip task activation volume was caused by changing brain organization rather than by changes in subject performance.

### 4.4. fNRIS Studies

#### 4.4.1. Liu et al., 2022 [46]

In this study, the efficacy of a BCI training neurorehabilitation protocol was assessed by using fNIRS centered on the activity of the primary motor cortex. The WMFT, and the FM subscores were employed to assess clinical changes after treatment. Eighteen hospitalized chronic stroke patients with moderate or severe motor deficits were enrolled. Functional evaluations were carried out at five points, defined as pre1-, pre2-, mid-, post-training, and 1-month follow-up.

The rehabilitation protocol was performed by a BCI-robot training system (RHB-III with 16 EEG channels; Shenzhen Rehab Medical Technology Co., Ltd., Shenzhe, China). It included a system for collecting real-time EEG signals, a control algorithm for central processing, and hand robot feedback. The paretic hand was fastened to the manus robotic exoskeleton in this system. As the tasks were being performed in a film, the subjects were encouraged to visualize doing the same thing with their paretic hand. A total of 160 trials were conducted during the BCI-robot therapy session across four runs of 40 trials each, with a 3 min rest in between. Each BCI training session lasted for roughly 40 to 50 min in total. There were 20 sessions in total.

The two stages of the experimental plan for the fNIRS testing were the resting state (RS) and task state (TS). The RS had to prepare the subjects, who were required to remain still for 10 s. The participants were asked to relax, close their eyes, and refrain from making any movements other than those necessary for the motor exercises in order to prevent manipulation of the blood oxygen data. In order to comprehend the instructions before the measurement, participants were instructed to practice the motor activity. For a total of 180 s, the participants were instructed to accurately execute a repeated three-time grip-and-rest exercise on a dynamometer.

The neuroplastic effects of BCI training revealed alterations in functional connectivity including the contralateral hemisphere in task-related brain activation induced by BCI training therapeutic intervention. Additionally, alterations in neuronal FC patterns were found in the ipsilesional motor and sensory regions. After BCI therapy, there was an increased pattern of connection between the ipsilesional primary motor cortex and the ipsilesional frontal cortex that accompanied the grasp of the damaged hand.

#### 4.4.2. Song et al., 2021 [47]

This RCT study looked at how robot-assisted gait training affected brain activity as measured by fNRIS. Thirty-six participants in all, eighteen in the Morning Walk group, and eighteen in the control group, were examined. Five times every week for three weeks for a total of fifteen times, were spent in rehabilitation. The Morning Walk group also received an additional hour of traditional physiotherapy in addition to 30 min of robot-assisted gait training every session. The control group, on the other hand, received conventional physiotherapy.

For the treatment of people with gait disorders, CUREXO Inc. in Korea developed Morning Walk, a lower limb end-effector robot, in 2014. It consists of a saddle that can support the patient’s weight and virtual reality training software that provides hands-on instruction. It may be used for free-walking and has numerous settings for walking speed, stride length, and other walking motions, including flat ground and stair-climbing [52]. Analyses were performed before the first therapy session and after the treatment program was completed.

Clinically, both groups’ FAC, MBI, BBS, and RMI scores significantly increased. The 10 MWT, gait speed, BBS, and MI-Lower scores significantly improved in the Morning Walk group compared with the control group.

Despite the authors not performing a direct comparison between groups, they described different neural changes occurring in the experimental groups. At the start of robot rehabilitation, the damaged hemisphere’s cortical activation was lower than that of the unaffected hemisphere. The damaged hemisphere experienced a greater increase in cortical activation following training, than the unaffected hemisphere. As a result, the damaged hemisphere’s cortical activation was noticeably higher than the unaffected hemispheres.

## 5. Discussion

Following a stroke, sensorimotor impairments are frequent and debilitating causes of overall functional disability. It is challenging to predict with any degree of accuracy how a stroke patient will eventually recover their ability to move. It would be highly useful clinically to find and characterize biomarkers that could predict either the course of recovery or response to specific therapies. This is particularly true in patients receiving robotics, given that no specific objective markers of recovery concerning advanced therapies are currently employed in clinical practice.

As previously suggested by Favre and colleagues [39], sensorimotor neural activity could be utilized to identify potential biomarkers of motor recovery since functional neuroimaging enables quick assessment of the neural mechanisms linked to stroke. In a quantitative coordinate-based activation likelihood estimation (ALE) meta-analysis of 24 neuroimaging papers assessing brain activity after rehabilitation, these authors consistently found greater ALE values in ipsilesional-MI and SMA of chronic stroke patients with good outcome when compared with those with poor outcome. Only trials using fMRI following conventional rehabilitation procedures were included. According to these authors, sensorimotor recovery is optimal when typical task-related activity patterns are re-established. Moreover, this meta-analysis showed ventral spatial shifts in ipsilesional task-related MI activity. It has been shown that the activation of sensorimotor representations that are not usually involved in hand function could be a sign of a vicarious recovery process that makes it easier to access on the direct corticospinal pathway’s intact sections [53,54]. This finding suggests that vicarious process reassignment can signify a successful adaptive restructuring of motor representations [55].

In this systematic review, regarding the functional biomarkers of robotic rehabilitation, we found that the premotor cortex, bilateral sensorimotor cortex, and ipsilateral M1 were the key neurofunctional changes observed in almost all studies. The vast majority of these neural reorganizations have been demonstrated to correlate with motor improvement. As concerns the presence of vicarious processes, there is a paucity of evidence about the recruitment of other cortical representations not typically and primarily devoted to hand/walking function.

The ipsilesional M1 is the main target for rehabilitation therapy because both animal and human studies have described neural plasticity changes in support to motor recovery of the controlateral affected side [56]. Patients with cortical strokes as well as those with subcortical strokes have posterior shifts in activation towards the postcentral gyrus, according to several neuroimaging studies examining the locations and extents of active areas in the ipsilesional primary sensory cortex [57,58]. After a robot-assisted virtual reality (RAVR) rehabilitation training, Saleh et al. [43] found an increasing relationship between the ipsilateral primary sensorimotor and motor cortices. In stroke victims, they found a correlation between increased sensorimotor processing associated with better motor performance. Following BCI-robot therapy, Liu et al. [46] described an increase in functional connectivity between the prefrontal cortex, ipsilesional M1, and controlateral M1 in the damaged hand’s grasp. This result supports evidence from a prior EEG study in which it was shown that, after BCI training, desynchronization over the ipsilesional central area during MI tasks is more pronounced than before the intervention, which suggests that BCI training has increased the motor area’s level of activity in the ipsilesional brain [59]. Therefore, there might be a neuroplastic response to the treatment between the motor system and the sensory system based on the variations in functional connection patterns during the grasp of the injured hand linked with the use of BCI therapy. Similarly, Moher et al. [40] using a BCI-guided robot-assisted (exoskeleton) system for upper limbs, described an increased functional connectivity of the ipsilesional primary motor cortex with the controlesional dorsal premotor cortex and SMA after training. Both these studies demonstrated the long-term effects of M1-based BCI-guided robot-assisted hand training in stroke rehabilitation. Indeed, while the entire training process was an active closed loop, the paretic hand’s movement is passively driven by the robot hand. Brain signals during motor imaging are used to decode the triggers for the robot hand, and the robot hand’s execution provides feedback on the users’ intended hand movements.

The intimate relationship between the M1 and the dorsal/medial premotor areas is another key neurophysiological substrate for supporting neural plasticity changes underlying successful motor recovery. Previous fMRI research has shown that the SMA is essential for motor recovery [60] at the chronic stage. This is in line with the evidence that SMA hyperactivity is associated with successful recovery both acutely and up to six months following stroke [61]. A longitudinal fMRI investigation found a favorable correlation between subsequent motor improvement and the retention of the ipsilateral M1 connection with the contralesional SMA [62]. The SMA may assist movement integration processing within the injured motor systems through its involvement in motor learning and in voluntary movement control [63,64]. Premotor areas’ altered connections with MI subregions may be reflected in changes to the hand’s somatotopic representations, which would help recruit premotor’s remaining corticospinal fibers and compensate for the corticospinal tract’s impaired monosynaptic projections. All this evidence has been confirmed by Moher et al. [40], who proposed that the integrity of the ipsilesional corticospinal tract was substantially correlated with both the improvement of motor function and the functional changes occurring between the ipsilesional M1 and contralesional premotor regions after robotic training. Although almost all studies have revealed an overall increase in bilateral premotor cortex activation, the variety of the motor tasks and imaging analysis prevents us from drawing a clear conclusion about the impact of neurorobotic rehabilitation [42].

Another point emerging from this review is the lack of evidence about the involvement of subcortical key regions involved in motor recovery. The cerebellum and basal ganglia are part of the primary motor pathway involved in every motor function. It has been reported that contralesional anterior cerebellar lobule activity declines early after stroke and is lower throughout the chronic period when compared with activity in healthy patients [39]. Additionally, the vermial activity is associated with poor recovery. In agreement with this evidence, Saleh et al. 2017 [43] found that the activity within the cerebellum was significantly reduced after RAVR-based training, but not in the RTP group. This controversial finding can be explained in light of monkey autoradiographic investigations, which show that the vermis sends bilateral projections to the thalamic nuclei, which serve as relays to the motor cortex [65]. Reduced vermial involvement might indicate the failure in the compensation for dentato-rubro-thalamocortical loop, thereby emerging as a possible limitation of the robotic device in inducing a complete motor recovery [39].

### Limitations

Currently, the determination of functional biomarkers associated with robotic rehabilitation is characterized by several limitations.

(1)A bias selection in studies included in this systematic review could be considered. The focus of this review was solely on the existence of stable neural plasticity changes associated with intensive robotic neurorehabilitation therapy. Due to this, we excluded a number of studies that looked at the neurological underpinnings of motor exercises performed with robotic devices and measured during the course of a single neuroimaging session;(2)Poor employment of the RCT study design. RCT study designs have not been extensively used in many studies in this field. Furthermore, only two of the few trials included in this review used a very active control condition that allowed separation of the impact of robotic interventions from other treatments;(3)The heterogeneity of the studies is the main weakness of this research. Indeed, individual patients’ data including delay after stroke, lesion site, clinical severity could not be effectively controlled across research. Additionally, neuroimaging task procedures, analysis, and type were also principal sources of variability.

Thus, considering points 1 and 2, it is plausible that there are few studies currently investigating the neural correlates of intensive robotic rehabilitation. Indeed, some challenges or practical considerations that may affect the feasibility of RCT in this field should be undertaken. For instance, neuroimaging studies are expensive and time-consuming, and structural connectivity imaging requires access to specialized technology and expertise. Given that only a few facilities can provide this type of diagnostic investigation, in literature we found only few and heterogeneous studies with small sample size. This is why, although the results of this review are promising, the use of fMRI to investigate functional recovery is far from being a standard tool in assessing patients with stroke.

## 6. Conclusions

Robots used in neurorehabilitation are designed to help administer physical therapy to the upper or lower extremities, with the goal of accelerating neuro-motor recovery. This technology dates back to the early 1990s. After more than 30 years, we do not know how robotic neurorehabilitation influences brain plasticity or functional brain rearrangement. This is no longer acceptable, in light of the growing development of more successful treatments (neural stimulators, BCI) and the growing acceptance of quality of life as the most important determinant of appropriate patient management. This information is mandatory for building and improving a new era of more effective devices with respect to traditional treatments.

In this systematic review, we sought to delineate the neurofunctional underpinnings of long-term neurorobotic effects on motor recovery in patients with stroke. Generally, a consistent cortical reorganization within the bilateral premotor cortex (dorsal/medial) and ipsilesional primary sensorimotor/motor cortices are indicative of successful motor recovery in a collection of data from seven distinct investigations of stroke recovery. According to previous evidence [39], we can postulate that advanced robotic rehabilitation may influence similar pathways to conventional therapy in improving functional recovery. In other words, it would seem that the neurofunctional correlates of successful motor recovery could be independent of treatment modality. However, the few studies included in this review as well as the other limitations, prevent us from generalizing the neurobiological basis of robotic-induced functional recovery to other neurorehabilitation approaches.

The neuronal remodeling underlying the primary motor cortex appears to be the only reliable data that might justify consideration as a potential biomarker for sensorimotor recovery in stroke patients after robotic treatments. Indeed, the most convincing evidence for a practical translation into clinical practice appears to be the M1-based BCI-guided robot-assisted training. Otherwise, the documented functional connectivity changes between the prefrontal/premotor areas and the motor cortex are intriguing but deserve further confirmations irrespectively from robotic device and motor tasks employed during fMRI/fNIRS investigations.

Finally, several developments were required before fMRI/fNIRS could be considered viable techniques for assessing the effectiveness of novel therapies designed to speed up stroke recovery. First and foremost, there is a need for more thorough RCT studies that compare robotic rehabilitation with conventional therapies while taking into account the neurofunctional rearrangements associated with the latter. To find out whether the neurofunctional patterns previously outlined have resulted in the substantial and long-lasting reshaping of the gray/white matter, structural connectivity imaging studies are next strongly recommended.

## Figures and Tables

**Figure 1 bioengineering-10-00990-f001:**
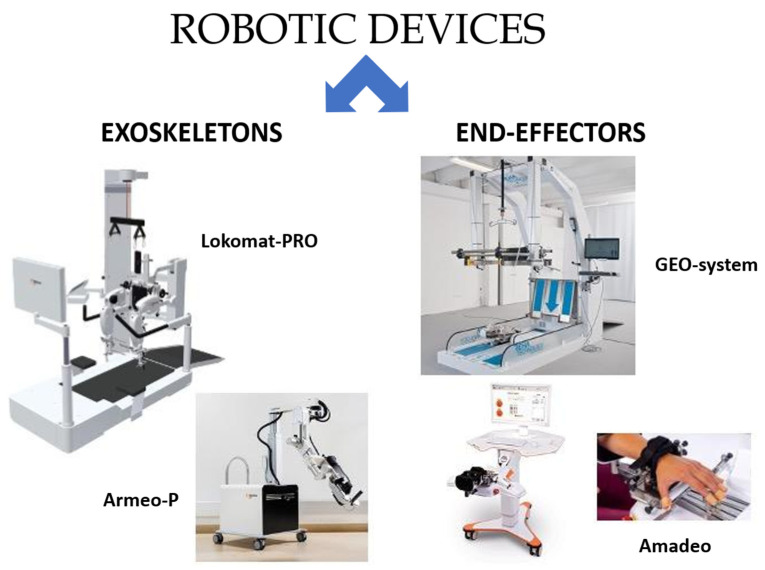
The most common exoskeletons and end effectors used in stroke rehabilitation. Lokomat^®^ and Armeo Power^®^ (Hocoma AG, Volketswil, Switzerland) for the lower and upper limbs, respectively. GEO-system^®^ (Reha Technology AG, Solothurnerstrasse 259, 4600 Olten, Switzerland) and Amadeo^®^ (Hobbs Rehabilitation, Winchester, Bridgets Lane, Winchester, UK) for lower and upper limbs, respectively.

**Figure 2 bioengineering-10-00990-f002:**
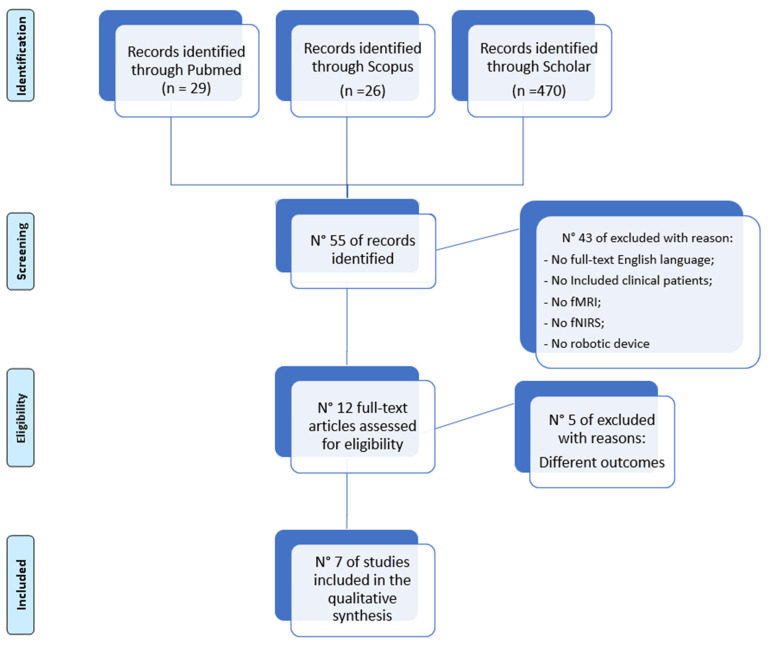
The PRISMA analysis.

**Figure 3 bioengineering-10-00990-f003:**
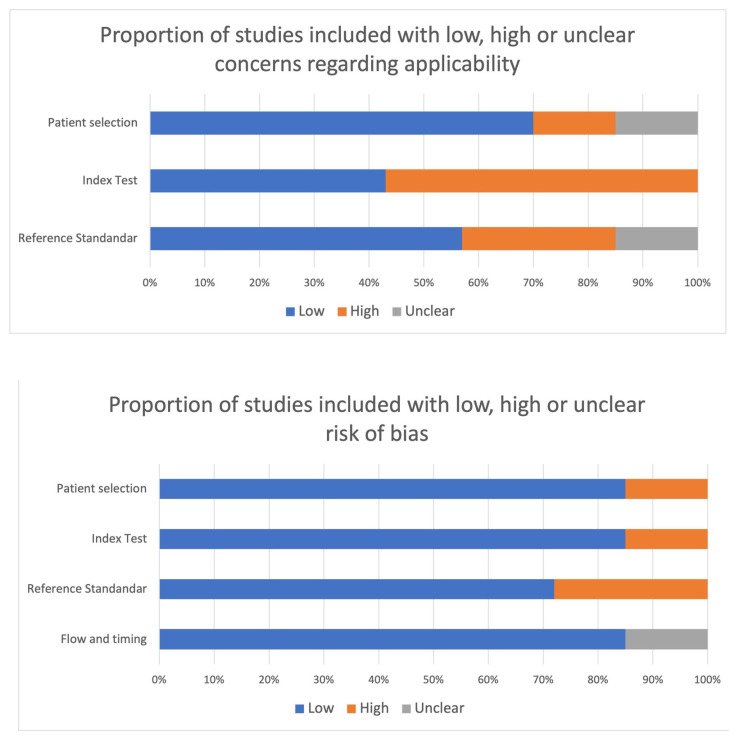
Results of the methodological quality and the risk of bias.

**Table 1 bioengineering-10-00990-t001:** Characteristics of studies applying functional neuroimaging to evaluate neural plasticity in stroke patients after robotic rehabilitation.

Reference	Neuroimaging Modality	Patients	Robotic Rehabilitation Procedure	Experimental Procedures	Clinical Outcome	Main Results
Yuan K. et al., 2020 [41]	fMRI	Stroke patient n° 14 (13 males, 1 female, age = 54 ± 8 y) with right (*n* = 9) or left (*n* = 5) lesions.	Subjects were asked to imagine either grasping or releasing a cup following the instruction displayed on the monitor. BCI-guided robot-assisted (exoskeleton) was used for upper-limb training. Patients were assisted for the paretic hand by an exoskeleton robot to carry out the grasping and opening operations. Treatment lasted for 20 sessions with an intensity of 3–5 per week and was completed within 5–7 weeks.	Resting state. fMRI motor task: imagery of grasping movements.	FMA	Robotic rehabilitation induced an increased functional connectivity between primary motor cortex and controlesional SMA and premotor cortex.
Astrakas L.G. et al., 2021 [42]	fMRI	Stroke patient n° 8 (4 males, 4 females, age = 49.9 ± 12.7 y). HC n° 13 (5 males, 8 females, age = 55.4 ± 13.1 y).	Patients performed a grip task at different force levels with the Magnetic Resonance Compatible Hand-Induced Robotic Device (MR_CHIROD) with an interactive computer game. Treatment lasted 45 min per day, 3 days per week, over a 10-week period.	fMRI motor task: compressing and releasing the MR_CHIROD synchronously (0.52 Hz) with a visual metronome cue. The HCs were submitted to a single scanning session, and stroke patients underwent five scan sessions.	FM UE MAS	In comparison with HC, patients moving the affected hand showed correspondingly higher peak activations in the primary motor area and lower peak activations in the somatosensory cortex. Additionally, they demonstrated an average 5.3 mm anterior shift in peak activity.
Saleh S. et al., 2017 [43]	fMRI	Experimental group (robot-assisted virtual reality, RAVR): stroke patient n° 10 (8 males, 2 females, age = 59.6 ± 10.6 y). Control group (repetitive task, RTP): stroke patient n° 9 (6 males, 3 females, age = 57.0 ± 12.8 y).	RAVR Group: Training involved reaching for and interacting with stationary and moving virtual targets and objects in 3D space playing with video games in a robot-assisted virtual reality setup (NJIT-RAVR), adapted to the individual needs of each subject. The training was conducted 3 h/day, 4 days/week, for 2 weeks. RTP Group: Training included 2 weeks of “reach and grasp” exercises and functional tasks of whole hand finger flexion with the paretic hand, and subjects were invited to flex their fingers to meet a set of two visual targets, angled at 40 and 80% of their maximum range of motion, in a VR simulation by using two MRI-compatible fiber-optic realized gloves.	fMRI motor task: hand finger flexion with the paretic hands.	JTHFT	A significant difference in lateralization index between groups was observed after training. The shift toward greater ipsilesional hemisphere dominance was more pronounced in the RAVR group than RTP group. A strong association between bilateral primary sensory regions, ventral premotor area, and ipsilateral primary motor cortex was found in the RAVR but not in the RTP group.
Saleh S. et al., 2012 [44]	fMRI	Robot-assisted virtual reality (RAVR), stroke patient (n° 2, male).	Patients performed RAVR training. This is an intensive upper extremity training protocol, where subjects play video games in a robot-assisted virtual reality setup (NJIT-RAVR). This setup works on retraining subjects’ hand/arm coordination, reaching, grasping, and finger individuation. The treatment lasted 3 h/day, 4 days/week, for 2 weeks.	Resting state. fMRI motor task: wearing a data glove on each hand to: (1) record the finger joint angle for offline analysis, and (2) stream these kinematics to a display in real-time thus providing subjects with online feedback of their movement via a virtual reality environment.	WMFT JTHF	Reorganization in functional connectivity between primary motor cortex and sensorimotor cortex and decreased connectivity between the contralesional sensorimotor cortex and primary motor cortex, while secondary sensorimotor cortex had substantial increase in connectivity between bilateral sensorimotor/premotor areas and primary motor cortex.
Takahashi et al., 2008 [45]	fMRI	Stroke patient n° 13 (6 males, 7 females, age age = 63 ± 16 y).	Patients underwent 9 cycles of 10 repetitions of simple grasp–release exercise together with exercises playing a set of interactive virtual reality computer games. Hand Wrist Assistive Rehabilitation Device (HWARD) was employed for the robotic therapy 15 daily sessions, on weekdays, over 3 weeks. Each session was 1.5 h long.	Passive fMRI task: viewing a guidance video that displayed the desired movement in the form of a stick-figure hand.	ARAT FMA NIHS S Geriatric Depression Scale Nottingham Sensory Assessment dynamometer recording of grip and pinch strength.	Increased activity within the left (stroke-affected) primary sensorimotor cortex after treatment.
Liu L. et al., 2022 [46]	fNIRS	Stroke patient n° 18 (14 males, 4 females, age = 45.33 ± 15.07 y).	BCI-robot training system consisting of a robotic exoskeleton to which the paretic hand was strapped. Participants were instructed to watch the actions displayed in the video and guided to imagine they were performing the same movement with the paretic hand. The treatment lasted 20 sessions (1 session per day, 5 days per week), and followed up after 1 month.	Resting state fMRI motor task: repetitive grip-and-rest task by using a dynamometer as accurately as possible.	WMFT FMA	Changes in neural functional connectivity were observed after BCI-robot training in both motor and sensory areas in the ipsilesional brain.
Song K.J. et al., 2021 [47]	fNIRS	Experimental group (Morning Walk group) n° 18 (12 males, 6 females). Control group n° 18 (9 males, 9 females).	The experimental group received robot-assisted gait training with the Morning Walk^®^ for 30 min per session, plus 1 h of conventional physiotherapy. The control group received 1.5 h of conventional physiotherapy, based on traditional neurodevelopmental treatment techniques developed by Bobath. The treatment was performed 5 times per week, for a total of 15 times in 3 weeks.	Cortical activation was measured by fNIRS during walking tasks performed at the first and last treatment sessions.	FAC 10 MWT RMI Motricity MI–Lower MBI	After robot-assisted rehabilitation, motor cortical activation was increased significantly in both hemispheres, and the degree of increased activation was greater in the affected hemisphere than in the unaffected hemisphere.

Legends: MAS (Modified Ashworth Scale); FMA (Fugl-Meyer Assessment); FM UE (Fugl-Meyer Upper Extremity); JTHFT (Jebsen Taylor Hand Function Test); WMFT (Wolf Motor Function Test); JTHF (Jebsen Test of Hand Function); RAVR (Robot-Assisted Virtual Rehabilitation); RTP (Repetitive Task Practice); ARAT (Action Research Arm Test); NIHSS (National Institutes of health Stroke Scale); FAC (Functional Ambulatory Category); 10 MWT (10 Meter Walk Test); RMI (Rivermead Mobility Index); MI (Motricity Index); MBI (Modified Barthel Index); fMRI (functional Magnetic Resonance Imaging); fNIRS (functional InfraRed Spectroscopy).

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
