# Peer review of "Neural Plasticity Changes Induced by Motor Robotic Rehabilitation in Stroke Patients: The Contribution of Functional Neuroimaging"

_bioengineering, 2023, doi:10.3390/bioengineering10080990_

Round 1
Reviewer 1 Report
An interesting systematic review investigating the role of robotic rehabilitation in neural plasticity in stroke patients. The study is performed according authors’ statement following the PRISMA guidelines, is well written and suitable for this special issue. The authors have searched numerous databases (PubMed, Psycho-info, PeDro, Scopus and Cochrane Library) to collect the body of evidence.
A few comments that may improve it.
1. In the search method it was not used the PICO approach according to PRISMA PICO is preferred, could the authors “rephrase” their query to be conformant with the PICO approach as required by PRISMA?
2. Figure 1 is of low quality, perhaps due to pdf conversion? The words are hardly readable, can be improved?
3. The quality of the manuscripts included in the review was not assessed by a method such as the RoB2 tool or the New Castle Ottawa approach or another suitable method as required by PRISMA. A table with studies quality and/or a traffic light plot could facilitate readers to assess the studies quality.
Author Response
An interesting systematic review investigating the role of robotic rehabilitation in neural plasticity in stroke patients. The study is performed according authors’ statement following the PRISMA guidelines, is well written and suitable for this special issue. The authors have searched numerous databases (PubMed, Psycho-info, PeDro, Scopus and Cochrane Library) to collect the body of evidence.
A few comments that may improve it.
- In the search method it was not used the PICO approach according to PRISMA PICO is preferred, could the authors “rephrase” their query to be conformant with the PICO approach as required by PRISMA?
REPLY: Done. Please see pag. 6
- Figure 1 is of low quality, perhaps due to pdf conversion? The words are hardly readable, can be improved?
REPLY: Figure 1 and Figure 2 have been improved.
- The quality of the manuscripts included in the review was not assessed by a method such as the RoB2 tool or the New Castle Ottawa approach or another suitable method as required by PRISMA. A table with studies quality and/or a traffic light plot could facilitate readers to assess the studies quality.
REPLY: In accordance with our previous systematic and meta-analytic studies (10.3390/brainsci11060727; 10.1016/j.neubiorev.2020.04.026) we now included the Quality Assessment of Diagnostic Accuracy Studies (QUADAS) evaluation.
Reviewer 2 Report
This article endeavors to assess the neurofunctional correlates of robotic therapies in enhancing motor recovery of stroke populations, a task of significant importance in the field of rehabilitation medicine. However, several critical weaknesses and limitations in the design, methodology, and interpretation of the study need to be addressed.
First, the study suffers from a small sample size, considering only seven out of the 150 initial articles for review. This low representation can drastically affect the validity and generalizability of the study's findings. It is important to explain why a large number of studies were excluded, as this could potentially introduce selection bias.
Second, the review highlights significant heterogeneity in robotic devices, clinical status, and neuroimaging procedures across the studies. However, it fails to thoroughly discuss how these disparities impact the review's findings and conclusions. In any systematic review, it is essential to discuss heterogeneity in depth as it can substantially influence the interpretation of the results.
Third, the absence of randomized control trial (RCT) studies in the review prevents the authors from making substantive conclusions regarding the neurobiological differences induced by robotic devices compared to traditional rehabilitation approaches. The lack of high-quality studies raises concerns about the review's quality and the overall robustness of the conclusions drawn.
Fourth, while the authors conclude that the neurofunctional correlates of successful motor recovery are independent of treatment modality, they provide limited evidence or justification for this claim. It would be beneficial if the authors had expanded on why they believe these findings are generalizable across different rehabilitation approaches, especially in light of the fact that their systematic review is focused solely on robotic rehabilitation.
Lastly, the article recommends several future research directions, such as conducting thorough RCTs and structural connectivity imaging studies. However, it fails to address potential challenges or practical considerations that may affect the feasibility of such research. For instance, RCTs are expensive and time-consuming, and structural connectivity imaging requires access to specialized technology and expertise. Recognizing and discussing these challenges would have grounded the article's recommendations in a more practical context.
In conclusion, while the article addresses a valuable area of research in stroke rehabilitation, it needs to critically address the limitations associated with the sample size, the heterogeneity of the reviewed studies, the lack of RCT studies, the substantiation of its claims, and the practical considerations for future research.
Author Response
This article endeavors to assess the neurofunctional correlates of robotic therapies in enhancing motor recovery of stroke populations, a task of significant importance in the field of rehabilitation medicine. However, several critical weaknesses and limitations in the design, methodology, and interpretation of the study need to be addressed.
- First, the study suffers from a small sample size, considering only seven out of the 150 initial articles for review. This low representation can drastically affect the validity and generalizability of the study's findings. It is important to explain why a large number of studies were excluded, as this could potentially introduce selection bias.
REPLY: We agree with the reviewer’s suggestion. A new section in the conclusions section has been included
2. Second, the review highlights significant heterogeneity in robotic devices, clinical status, and neuroimaging procedures across the studies. However, it fails to thoroughly discuss how these disparities impact the review's findings and conclusions. In any systematic review, it is essential to discuss heterogeneity in depth as it can substantially influence the interpretation of the results.
REPLY: We completely agree with this reviewer. Generally, in our previous meta-analytic studies (i.e., 10.1016/j.neubiorev.2020.04.026), we used the heterogeneity I2 index as follows:
I2= [(χ2 – df)/ χ2]*100.
However, this metric makes sense when a large number of studies are considered. In this case, with a limited number of papers, this kind of metric could be quite useless. We specify many times in the text that the main limitation of this study is the presence of high heterogeneity in study design, fMRI methods and robotic approaches.
We are aware that this could be a great limitation of this study. However, on the other hand, the importance of this review is to launch an alert about the fact that:
“Robots used in neurorehabilitation are made to help administer physical therapy to the upper or lower extremities with the goal of accelerating neuro-motor recovery. This technology dates back to the early 1990s. After more than 30 years, we don't know how robotic neurorehabilitation influences brain plasticity or functional brain rearrangement. This is no longer acceptable, in light of the growing development of more successful treatments (neural stimulators, BCI) and the growing acceptance of quality of life as the most important determinant of appropriate patient management. This information is mandatory for building and improving a new era of more effective devices with respect to traditional treatments”. [See Conclusions Section]
For this reason, we highlight the importance of our systematic review which could serve as a starting point for further investigations and inspire advancements and clinical studies in the field. Exactly how it happened for our other previous review (10.1177/1545968317693304)
3. Third, the absence of randomized control trial (RCT) studies in the review prevents the authors from making substantive conclusions regarding the neurobiological differences induced by robotic devices compared to traditional rehabilitation approaches. The lack of high-quality studies raises concerns about the review's quality and the overall robustness of the conclusions drawn.
REPLY: We agree with the reviewer’s suggestion. A new section in the conclusions section has been included including Question n°1.
4. Fourth, while the authors conclude that the neurofunctional correlates of successful motor recovery are independent of treatment modality, they provide limited evidence or justification for this claim. It would be beneficial if the authors had expanded on why they believe these findings are generalizable across different rehabilitation approaches, especially in light of the fact that their systematic review is focused solely on robotic rehabilitation.
REPLY: The reviewer is right in that we do not have evidence to claim that these findings are generalizable, considering the few studies included and the small sample size. We have added this concern and tuned down the statement.
5. Lastly, the article recommends several future research directions, such as conducting thorough RCTs and structural connectivity imaging studies. However, it fails to address potential challenges or practical considerations that may affect the feasibility of such research. For instance, RCTs are expensive and time-consuming, and structural connectivity imaging requires access to specialized technology and expertise. Recognizing and discussing these challenges would have grounded the article's recommendations in a more practical context.
REPLY: As suggested, we added the limitations and challenges of applying these methods for both future research and clinical practice.
6. In conclusion, while the article addresses a valuable area of research in stroke rehabilitation, it needs to critically address the limitations associated with the sample size, the heterogeneity of the reviewed studies, the lack of RCT studies, the substantiation of its claims, and the practical considerations for future research.
REPLY: These further limitations have been critically added, as suggested
Round 2
Reviewer 2 Report
There are several issues that must be addressed in a revised version of the manuscript:
· Figure 1: provide sources or references for the discussed devices.
· Explain in more detail, how QUADAS was applied.
· Explain how records were merged from the three bibliographic datasets.
· The discussion on bias is too short and insufficient.
· The contribution of this review study over previous reviews is not clear. Discuss added knowledge over related review studies such as “Application of Internet of Things and Sensors in Healthcare”, “A review of internet of things technologies for ambient assisted living environments”, “IoT-Based Applications in Healthcare Devices”.
· Discuss knowledge gaps covered by the current review.
Author Response
There are several issues that must be addressed in a revised version of the manuscript:
- Figure 1: provide sources or references for the discussed devices.
Reply: Done
- Explain in more detail, how QUADAS was applied.
Reply: A new figure reporting the results of QUADAS analysis has been included (Figure 3).
- Explain how records were merged from the three bibliographic datasets
Reply: Done
- The discussion on bias is too short and insufficient.
Reply: We disagree with this reviewer because the main bias and limitations of this field of study have been already completely described.
- The contribution of this review study over previous reviews is not clear. Discuss added knowledge over related review studies such as “Application of Internet of Things and Sensors in Healthcare”, “A review of internet of things technologies for ambient assisted living environments”, “IoT-Based Applications in Healthcare Devices”.
Reply: The contribution of this review is to alert bioengineers, neuroscientists, and neurologists that after more than 30 years, we don't know how robotic neurorehabilitation influences brain plasticity or functional brain rearrangement. We believe that this first attempt to summarize the evidence in this field of study is of crucial interest for future studies.
- Discuss knowledge gaps covered by the current review
Reply: The gap covered by this review, has been already discussed. The starting point for future studies defining biomarkers of robotic neurorehabilitation should begin with M1-based BCI-guided robot-assisted training. These studies currently offer the most solid and trustworthy evidence.
Round 3
Reviewer 2 Report
The reviewers have addressed all my comments and revised the manuscript accordingly. The manuscript can be accepted for publication.